# The activity of soil microbial taxa in the rhizosphere predicts the success of root colonization

Jennifer E. Harris,[1,2,3] Regina B. Bledsoe,[1] Sohini Guha,[1] Haneen Omari,[2] Sharifa G. Crandall,[4] Liana T. Burghardt,[1,2] Estelle Couradeau[2,3]

**ABSTRACT** Plant-beneficial microbes have great potential to improve sustainability in agriculture. Still, managing beneficial microbes is challenging because the impact of microbial dormancy on community assembly across the soil, rhizosphere, and endosphere is poorly understood. We address this gap with the first documented use of Biorthogonal Non-Canonical Amino Acid Tagging (BONCAT) to probe active microbes in the soil-to-root gradient. Using nodule-forming legume *Trifolium incarnatum,* we confirmed that BONCAT is suitable for labeling endospheric microbes with microscopy. Next, we coupled BONCAT to Flow Cytometer Activated Cell Sorting (FACS) and 16S rRNA amplicon sequencing to probe patterns of microbial activity and the structure of the active microbial community across the soil, rhizosphere, root, and nodule with a native soil microbial community. As expected, we found 10 times higher microbial activity in the endosphere than in the rhizosphere or bulk soil, likely due to increased plant resources. Finally, we revealed that microbial activity in the rhizosphere was significantly associated with successful endosphere colonization, more so than microbial abundance alone. This last finding has implications for the development of microbial inoculants, suggesting colonizing plant roots is linked to a microbe's ability to overcome dormancy once deployed in the soil.

**IMPORTANCE** Most soil microbes are dormant, so they must exit dormancy to have the potential to carry out plant-beneficial functions. It is unclear if dormant microbes revive in proximity to plant-produced resources and if overcoming dormancy in the soil is important for successful plant colonization. We use a novel microbial activity probing technique for the first time on and in plant roots, and with it, we observe microbes increased in activity 10× inside plant tissues compared to the soil, likely in response to plant-produced resources. In complex, native microbial communities, we observe that microbes that are active and abundant are more likely to colonize plant roots successfully than just abundant microbes. Our research shows that plants could be leveraged to promote a distinct active microbial community from the native soil, a discovery that has the potential to improve sustainability in agriculture.

**KEYWORDS** microbial activity, microbial dormancy, plant-microbe interactions, 16S RNA, BONCAT, microbial ecology, environmental microbiology, soil microbiology

Beneficial microbes could enhance the sustainability and resiliency of agriculture (1, 2). Microbes can promote nutrient delivery to plants, increase plants' tolerance to drought stress, and compete with soil-borne pathogens (3–7). These benefits can decrease farmers' dependency on inorganic fertilizers, allow plants to persist in a changing climate, and decrease agriculture's reliance on toxic pesticides (7–9). Populations of beneficial microbes could be promoted directly with microbial inoculants or indirectly with cover crops (8). Microbial inoculants deliver a microbial product with a

Address correspondence to Estelle Couradeau, estelle@psu.edu.

Liana T. Burghardt and Estelle Couradeau contributed equally to this article.

The authors declare no conflict of interest.

See the funding table on p. 16.

predictable function in the laboratory, but they often fail to establish the field because they cannot compete with the native microbial communities (8, 10). A way to overcome this challenge would be to leverage existing dependencies between cover crops and beneficial microbes to manage microbes from the native soil community indirectly (11–14). Native microbes could more effectively improve plant health than non-native microbes because they are locally adapted to the environment (15). However, so far, we cannot predictably increase relevant microbial functions or taxa with cover crops (16). Failure of both direct and indirect microbial management is likely because more than 90% of microbes, including beneficial ones, remain dormant in the soil (17). We do not know if plants can revive dormant beneficial microbes from the soil community or if microbes must overcome dormancy in the soil for successful plant colonization. To promote beneficial microbes with indirect or direct management, a better understanding of microbial activity and establishment in environments from the soil to the roots is needed. We address this gap by tracing the active microbial community across the soil, rhizosphere, endosphere, and nodule in a legume cover crop.

In the spatial gradient from soil to roots, microbes can experience massive shifts in environmental conditions over just a few millimeters. The bulk soil, characterized by diverse, sparsely distributed resources, supports many microbial niches (18) and immense microbial diversity (19). In the soil directly adjacent to the plant root, the rhizosphere, microbes are exposed to root exudates. These plant-produced compounds deter some microbes and promote others, often creating a distinct microbial community in the rhizosphere compared to the bulk soil (17, 18, 20, 21). From the rhizosphere to the root endosphere, microbes encounter a new suite of plant-produced resources and stresses, and microbial diversity decreases (21, 22). Legume nodules are an extreme example of ecological filtering of the microbial community. This spatially distinct root compartment is near-exclusively inhabited by rhizobia, a group of microbes that fix atmospheric nitrogen for the plant in exchange for carbon-rich photosynthates (23). Overall, some microbes can transition into new habitats across the soil, rhizosphere, and endosphere, and others cannot (22). It is unclear if the ecological filtering effect observed in total microbial DNA occurs in active microbial communities or is driven by dormant taxa (21, 22, 24, 25).

Distinguishing metabolically active microbes from dormant ones is important because active microbes are more functionally relevant across the soil-to-root gradient. In the soil, more than 90% of microbes are dormant (17), yet active microbes disproportionally affect soil physiology and emergent properties (26, 27). Dormant microbes can persist in resource-limited conditions and exit dormancy when conditions improve (28). Plant resources in the rhizosphere could revive microbes from dormancy and create a hotspot of microbial activity and biomass (29). We observe if microbes respond to plant resources by tracking microbial activity across the gradient of plant influence from the soil to the roots. We expected active soil microbes to be more likely to colonize plant roots (30). In the model *Sinorhizobium-Medicago* system, *Sinorhizobium* must be active to respond to plant signaling molecules and initiate nodule formation (30). However, most beneficial microbes do not have the same unique co-evolutionary history with their plant host as nodule-forming Sinorhizobium, so it is unknown if they would also require soil activity to colonize the rhizosphere successfully. Many techniques can probe the active microbial community, like stable isotope probing (SIP) (31, 32), transcriptomics (33), or Bioorthogonal Non-Canonical Amino Acid Tagging (BONCAT) (34). BONCAT is relatively affordable to implement and does not require cell growth for labeling, as it relies on labeling newly synthesized proteins (35, 36). Furthermore, BONCAT is a robust and effective microbial activity probe that is comparable to other activity metrics in environmental samples (37–41) and in bulk soil (38). For these reasons, we implement BONCAT for the first time in living plant tissue. Doing so will allow us to observe microbial establishment and activity across this plant-soil resource gradient with the long-term goal of promoting agriculturally relevant microbes.

To better understand the establishment of microbial taxa across the soil-root gradient, we ask, (i) do plant resources increase microbial activity, (ii) which microbial taxa are active across the soil-root gradient, and (iii) is the microbial activity of taxa in the rhizosphere associated with successful root colonization? In this study, we deployed BONCAT-FACS-Seq in a spatially resolved manner in the soil-plant continuum, sampling four compartments: the soil, the rhizosphere, the root endosphere, and the nodule in legume cover crop *Trifolium incarnatum* with a native soil microbial community. We visualized active microbes with fluorescence microscopy and measured overall microbial activity across these compartments with flow cytometry. To identify active microbes across compartments, we use BONCAT coupled with fluorescence-activated cell sorting (FACS) and 16S rRNA amplicon sequencing (38). We performed a differential abundance analysis comparing the active vs viable taxa in the rhizosphere at the phylum and amplicon sequence variant (ASV) levels. Finally, we examined patterns of plant endosphere colonization, as defined by microbial presence in plant structures. For the first time, we demonstrate BONCAT as an alternative technique to measure patterns of microbial activity in the rhizosphere and endosphere. With this technique, we pinpoint disproportionately active taxa in the rhizosphere, zeroing in on taxa with the potential to drive beneficial functions in a native microbial community. Finally, we show that microbial activity in the rhizosphere is significantly associated with successful microbial plant tissue colonization and discuss its implications for microbial management.

## MATERIALS AND METHODS

### Greenhouse experiment

In a greenhouse, we grew *Trifolium incarnatum* in a 50:50 mixture of unsterile field soil and sand (650 mL Dee-Pots, nine replicates, and four soil-only controls) in a randomized block design. We collected field soil from the Russell E. Larson Agricultural Research Center at Rock Springs, PA (40.7215, −77.9275) (42), which has a history of *Trifolium incarnatum* cultivation, so our greenhouse plants were exposed to a natural population of plant-associating bacteria. After 8 weeks of growth, we dosed six plants with 100 mL of aqueous 250 µM L-Homopropargylglycine (HPG, 0.038 µmol per gram of soil), one plant with 250 µM aqueous methionine, and two plants with water only. LCMS confirmed that our dosage level of HPG was higher than the natural methionine levels in the soil (Fig. S1, methods in the supplemental material). Previous work described in references 35 and 38 determined that the uptake of HPG in soil peaks in 24 h. We piloted longer incubation times because we suspected agricultural soil would have slower turnover rates but determined that 24 h also had the highest labeling for our soils (Fig. S2). After incubation, we sampled four compartments (bulk soil, rhizosphere, root endosphere, and nodule, Fig. 1). To collect bulk soil, we took a 6 mm diameter core to a depth of 5 cm in each pot. We added 2 g of the bulk soil to 10 mL of phosphate-buffered saline with 0.02% Tween 20 (PBS-Tween) and vortexed for 1 minute to wash cells off soil particles. To collect the rhizosphere, we gently shook the root system to remove bulk soil, added the roots to 30 mL of PBS-Tween, vortexed the root system for one minute, and decanted the rhizosphere slurry. We picked all nodules from the root system and separately sterilized nodules and roots in bleach for 30 s and rinsed them in DI water for 1 min. Separately, we ground nodules and roots for 90 s in 10 mL of PBS-Tween with a tissue homogenizer (7 mm probe, Omni International) to disrupt the plant tissue and release endophytes. Both sample types were centrifuged at 300×*g* for 3 min to pellet large debris. We stored the supernatant, enriched for endophytes from each compartment (bulk soil, rhizosphere, root endosphere, and nodule), in 1 mL aliquots in 10% glycerol at −20°C for future processing.

### Probing microbial activity with BONCAT and flow cytometry

We used BONCAT to label metabolically active microbial cells in all compartments (bulk soil, rhizosphere, root endosphere, and nodule). During incubation, active microbes

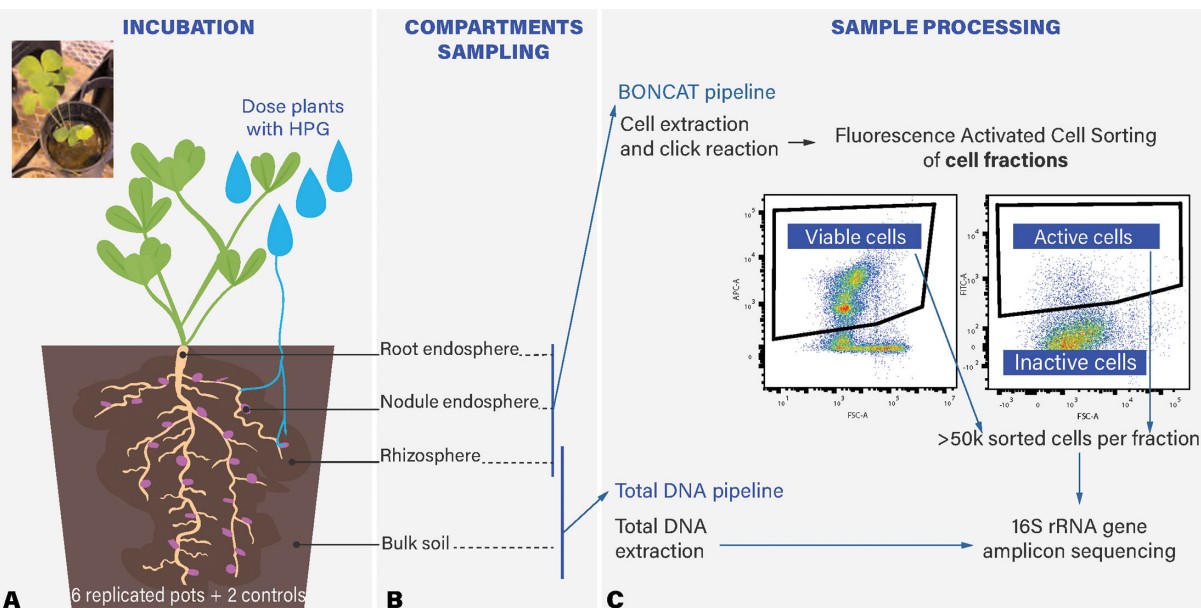

**FIG 1** Experimental design: probing microbial activity and composition across a gradient of plant-soil influence. (A) Six replicates of *Trifolium incarnatum* were grown in a greenhouse. After 8 weeks, plants received HPG in a simulated rain event. (B) After a 24 h incubation, we sampled four compartments: root endosphere, nodule endosphere, rhizosphere, and bulk soil. We measured activity in all compartments as the % of BONCAT fluorescence cells over the total number of cells and (C) established pipelines to measure microbial community composition using 16S RNA sequencing from (i) total extractable DNA from bulk soil and rhizosphere and (ii) active and viable fractions identified using BONCAT-FACS-Seq on three compartments (rhizosphere, root endosphere, and nodule endosphere).

incorporated a methionine analog (L-homopropargylglycine [HPG]) into newly made proteins. After incubation, a copper-catalyzed click reaction attached the HPG to a fluorescent probe (FAM picolyl azide dye, Click Chemistry Tools, Ex/Em 490/510 nm), making all translationally active cells fluorescent (38). We measured each compartment's portion of active cells to viable cells on a flow cytometer (BD LSRFortessa Cell Analyzer). To identify viable cells from soil particles, broken cells, and free DNA, we used SYTO59 DNA counterstain (concentration 0.5 µM, Invitrogen, Ex/Em 622/645 nm). We identified active cells as SYTO59-positive and BONCAT FAM picolyl azide-positive against an HPG-free water-incubated clicked control, so we could confirm the signal was from our HPG probe, not background noise. Then, we coupled BONCAT to fluorescence-assisted cell sorting (FACS) to collect active and viable cells from three compartments: rhizosphere, root endosphere, and nodule. We used the same gating strategy to identify and collect active cells from each sample with FACS on a MoFlo Astrios Cell Sorter (Beckman Coulter). To collect viable cells with FACS, we use SYBR Green positive gated against an unclicked unstained control (SYBR concentration 0.5 µM, Invitrogen, Ex/Em 498/522 nm). To determine cell density, we counted the number of SYBR Green positive events per microliters on a MACSquant VYB flow cytometer (Miltenyi Biotec). We collected at least 50,000 active and 100,000 viable cells from each sample. Our BONCAT FACS pipeline resulted in percentage microbial activity values for all four compartments and active and viable cell fractions from three compartments (rhizosphere, root endosphere, and nodule). See the supplemental material for additional information on click reaction, laser, and gating.

## DNA extractions and 16S rRNA amplicon sequencing

We sequenced paired active and viable fractions from three compartments (rhizosphere, root endosphere, and nodule from four replicate pots). We also sequenced total DNA from the soil from plant-free controls, bulk soil, and rhizosphere (four replicate pots). We adapted the library prep protocol from (39) for cell fractions. Cells were lysed with

prepGEM Bacteria Kit (New England Biolabs). We extracted total DNA from bulk soil and rhizosphere with the DNeasy PowerSoil Pro Kit (Qiagen). We used 515f-Y (30) and 806R-B (31) primers to amplify the 16S rRNA gene from lysed cells and total extractable DNA. The Huck Life Science Genomics Core carried out library preparation, indexed, sequenced, and demultiplexed libraries on an Illumina MiSeq platform with 250 bp × 250 bp paired-end sequences. The sequencing run produced, on average, 107,255 paired reads per sample. Additional PCR methods are in the supplemental material.

## Processing of 16S rRNA gene amplicon-sequencing data

We analyzed the demultiplexed 16S rRNA gene amplicon data with the QIIME two pipeline (v2022.2) (43), trimming forward and reverse reads at 220 base pairs (bp) where Phred scores dropped below 30. We paired forward and reverse reads, denoised sequences, removed chimeras, and assigned amplicon sequence variants (ASVs) with Divisive Amplicon Denoising Algorithm 2 (DADA2) pipeline (44) in Qiime2 (43). In DADA2, we set expected errors to 2 bp forward and 5 bp reverse, detection method to consensus ("maxEE = 2, 5, method = consensus"), and all other parameters to defaults. We used the Qiime2 "feature-classifier" to assign taxonomy, with the Naïve Bayes taxonomic classifier pre-trained on the Greengenes2 database (45). Unclassified ASVs in the roots and nodules were screened for plant contamination with NCBI BLAST (46). ASVs belonging to chloroplasts or mitochondria were removed with R package Phyloseq, version 1.44.0, with the "subset_taxa" function. This quality-controlled data set was rarefied with the "rrarefy" function in the R Vegan package, version 2.6-4 (38), to the minimum depth of 41,620 reads per sample. To compare the active and viable fractions with less noise from extremely rare taxa, we used the R package MicEco, version 0.9.19 (47), to filter the rarefied data set to only include ASVs (i) present in at least three samples in the active or viable fractions and (ii) with more than 50 reads in the active or viable fraction. This resulted in three datasets: the unrarefied data set, the rarefied data set, and the rarefied, filtered data set.

## Statistics of microbial diversity and community composition

We carried out statistical analysis in R, version 4.3.1 (48). Throughout the statistical analysis, compartment refers to bulk soil, rhizosphere, root endosphere, and nodule; fraction refers to total DNA, active, viable; and replicate pot refers to the blocking in our design. We used the rarefied data set and linear models to test for shifts in three alpha diversity metrics (richness, evenness, and Shannon diversity) calculated with the R package Phyloseq, version 1.44.0 (49). After finding a significant fraction-compartment interaction for all diversity metrics (Tables S3 to 5), we reran the model on each fraction separately to test for shifts in diversity between compartments. We also used the rarefied data set to test if microbial community composition changes across fractions and compartments. We calculated Bray-Curtis distances between samples (vegan, version 2.6-4 [50]) and visualized these distances using a PCoA that included computing 95% confidence intervals on sample groups. We calculated the beta dispersion of groups (vegan function "betadispr") and tested for homogenous variance with a permutation test (vegan function "permutest"). We used permutational multivariate analysis of variance (PERMANOVA) using distance matrices to assess if microbial communities differed significantly among compartments and fractions (Tables S6 and S7).

To test whether microbial taxa that were abundant in the rhizosphere viable fraction also tended to be abundant in the rhizosphere active fraction, we used a log-transformed linear model with a fixed effect of replicate pot with the rarefied, filtered data set (*log(abundance active)~log(abundance viable)+replicate pot*). Including the replicate pot in our model allows us to account for environmental variation in the greenhouse. With the same data set, we determined which phyla were enriched or depleted in the active fraction compared to the viable fraction of the rhizosphere with a separate generalized linear model with a binomial link for each of the six most abundant phyla. The dependent variable was the number of reads belonging to the focal phylum vs remaining reads.

We tested for the effect of fraction while controlling for the replicate pot (*Proteobacteria reads: remaining reads ~fraction + replicate pot*). We identified ASVs enriched or depleted in the active fraction relative to the viable fraction with differential abundance analysis from the ANCOMBC package in R version 2.2.2 (51) (alpha = 0.001, converse = TRUE, verbose = TRUE, all other parameters set to default) with the rarefied, filtered data set.

We tested for the association between the abundance of taxa in the rhizosphere viable fraction or the rhizosphere active fraction and successful plant colonization with a binomial model with a logit link. To avoid using correlated predictors in a model, we use AIC-based selection of binomial models to ask if the abundance of taxa in the rhizosphere viable fraction or the rhizosphere active fraction better predicts successful root colonization. We used the rarefied, filtered data set and defined successful plant colonization as an ASV with at least 50 reads across all root or nodule endosphere compartments in the active or viable fraction. We compared the AIC of our null model (*successful plant colonization(0,1) ~ replicate pot*) to a model that also included the abundance of taxa in the viable rhizosphere or the abundance of taxa in the active rhizosphere. We compared the AIC values across these three models, with lower AIC values indicating a better fit.

## Flow cytometry statistics

A quasibinomial model was used to compare the portion of active cells to viable cells across compartments. This model effectively modeled the distribution of proportion data and accommodated the overdispersion found in the data set. A likelihood ratio test with a χ distribution was used to test differences between compartments. The final model was *BONCAT active cells viable cells ~ compartment*.

## Microscopy

We used confocal microscopy to visualize BONCAT-labeled active microbial cells in the nodule. Pre-inoculated *Trifolium incarnatum* was grown in potting soil in the greenhouse. After 8 weeks, three plants were dosed with HPG dissolved in water at a final concentration of 0.038 µmol HPG per g soil, and one plant with HPG-free water as a negative control. After 24 h, nodules were picked and cut into 50 µm thick sections from the distal to the proximal end with Leica CM1950 Cryostat. The BONCAT reaction mix was the same as our flow cytometry samples. However, we substituted Picolyl-Azide-5/6-FAM for Cy3 Picolyl Azide (Click Chemistry Tools, Ex/Em 553/568, supplemental material). BONCAT labeled sections were imaged on the Leica SP8 DIVE multiphoton microscope. For visualizing nodule structure, sections were incubated for 5 min with 0.5% toluidine blue. Toluidine blue images were captured on a Zeiss Axio Observer fluorescence microscope with an Axio 208 color camera. Images were processed using ImageJ (Fiji) (52). Additional methods are in the supplemental material.

## RESULTS

### Flow cytometry and microscopy confirm the successful labeling of active bacteria with BONCAT in the soil-rhizosphere-root-nodule compartments

BONCAT coupled flow cytometry indicates labeling metabolically active taxa in all compartments (Fig. 1, Fig. S1). Because it was the first time that the technique was used *in planta*, we performed BONCAT coupled to fluorescence microscopy on intact *Trifolium incarnatum* nodule sections. BONCAT fluorescence is detected *in situ* in HPG-incubated sections (Fig. 2A, red) compared with the HPG-free (water-incubated) control where the fluorescence is absent (Fig. 2B, red). Endosymbiotic bacteria were BONCAT labeled and identified in the nodule by their compartment, size, and shape. Active bacteria are concentrated in the infection zone of the nodule (Fig. 2D). Bacteria are in regular groupings indicative of a peri-bacteroid membrane (Fig. 2E). Moreover, bacteria are rod-shaped and between 1 and 5 µm, consistent with nodule-inhabiting rhizobia (Fig.

2F). Plant cells in the nodule and root are also BONCAT labeled, indicating that some of the HPG was taken up by the plant. Plant cells were identified by their shape, size, and ability to be stained with toluidine blue dye (Fig. 2D).

## Microbial activity increases as diversity decreases across the rhizosphere, root endosphere, and nodule

We expected microbial activity to increase across the bulk soil, rhizosphere, root, and nodule due to increasing access to plant-produced resources near the root. Microbial activity, measured as the proportion of active cells to viable cells, is not significantly different between the bulk soil and rhizosphere (Fig. 3A). However, microbial cell density (cells/µL) is higher in the rhizosphere than in bulk soil ( Fig. S4, $P < 0.001$, df = 16), which suggests a greater number of active cells in the rhizosphere than in bulk soil. Microbial activity increases from the rhizosphere into the root endosphere and nodule ($P < 0.001$, df = 10 and $P < 0.001$, df = 10, respectively, Fig. 3, Table S1), which suggests that the plant interior has more dense microbial resources than the rhizosphere. Interestingly, microbial activity in both nodules and roots peaked around 10% of viable cells, which was contrary to our initial expectation that nearly all microbes would be active in the plant, suggesting that the plant environment could be stressful for microbes.

As expected, microbial diversity decreases across the bulk soil, rhizosphere, root endosphere, and nodule endosphere (Fig. 3B and C, Tables S3 to S5). Within the total DNA fraction, Shannon diversity decreases from the bulk soil to the rhizosphere ($P < 0.001$, df = 11). Shannon diversity decreases from the rhizosphere to roots for active and viable fractions ($P < 0.001$, df = 6 and $P < 0.001$, df = 7, respectively). Finally, Shannon diversity decreases from the roots to nodules in the active fraction ($P = 0.001$, df = 7). Across fractions, less than 10% of taxa in the rhizosphere are present in the plant, and less than 3% of taxa in the rhizosphere make it into the nodule. We observe the same pattern in unrarefied and rarefied data (Fig. S6). This decrease in Shannon diversity and the number of ASVs indicates ecological filtering across the rhizosphere, root endosphere, and nodule.

We expected the alpha diversity to be higher in the viable fraction than in the active fraction. In the rhizosphere and nodule, the decrease in Shannon diversity from the viable to the active fraction was not significant. In the root endosphere, Shannon diversity is marginally higher in the active compared with the viable fraction (ß = 0.15, $P$ = 0.04, df = 6). This unexpected result in the root endosphere could be due to our sorting method. We sorted at least 50,000 active cells labeled with SYTO 59 + BONCAT azide dye and 100,000 viable cells labeled with SYBR. It is possible that SYBR-labeled plant debris caused the true number of cells sorted to be lower than 100,000, preventing the detection of some rare ASVs in the root endosphere.

## Active and viable microbial communities are distinct across the rhizosphere, root endosphere, and nodule

Microbial beta diversity changes across the bulk soil, rhizosphere, root endosphere, and nodule (Fig. 4A). Regardless of fraction (active, viable, or total DNA), bulk soil and rhizosphere compartments have a nonoverlapping 95% confidence interval (CI) in the PCoA ordination of Bray-Curtis dissimilarities (Fig. 4A). The plant compartments (root endosphere and nodule) cluster separately from rhizosphere and bulk soils and are also significantly different from each other (PERMANOVA, root endosphere vs nodule, permutations = 999, $R^2$ = 0.37, $P$ = 0.007, df = 16). Active and viable fractions are significantly different in all compartments. The active and viable fraction's 95% confidence intervals did not overlap in the rhizosphere. However, a PERMANOVA was unsuitable to test this difference because of the nonhomogeneous variance (Beta dispersion test, $P$ = 0.001, df = 7). In the root endosphere and nodule, active and viable fractions were significantly different (PERMANOVA, $P$ = 0.024, $R^2$ = 0.60, df = 7, and $P$ = 0.022, $R^2$ = 0.51, df = 8, respectively). Full beta dispersion and PERMANOVA results are in

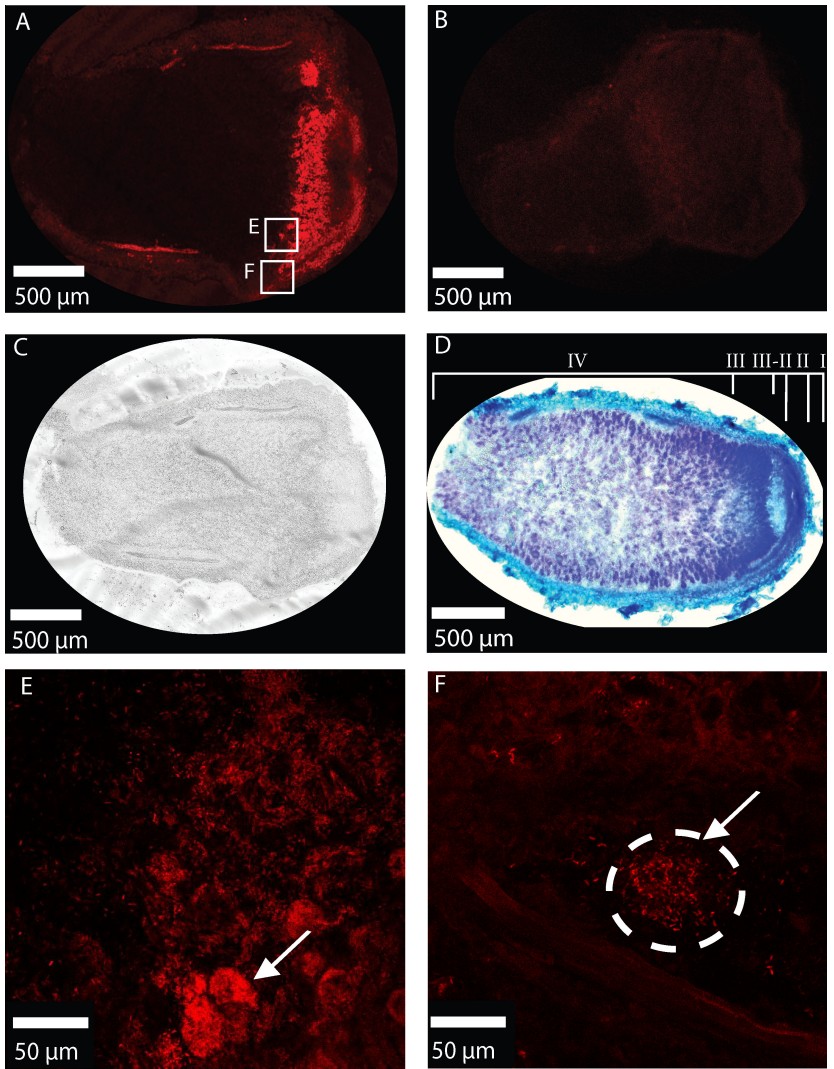

**FIG 2** Visualizing metabolically active endophytes with BONCAT. All images are from longitudinal sections of nodules of 8-week-old *trifolium incarnatum*. (A) HPG-incubated nodule with red BONCAT signal. Squares indicate zoomed-in areas in panels E and F. (B) HPG-free nodule imaged under the same condition lacks BONCAT signal. (C) Brightfield image of HPG-incubated nodule. (D) Toluidine blue stained nodule. Nodule zones labeled starting from the distal end to proximal to the root: (I) meristem (II) infection and differentiation zone (II–III) intermediate zone (III) nitrogen fixation zone (IV) senescence zone. (E) BONCAT labeled cluster of Rhizobium cells. (F) BONCAT labeled Rhizobium. Panel D was collected on a Zeiss Axio Observer fluorescence microscope, with an Axio 208 color camera. Panels A, B, C, E, F, and G were collected by Leica Dive M5 confocal microscope on a diode 553 nm laser line.

Tables S6 and S7. We observe the same pattern in unrarefied and rarefied data (Fig. S7). These results indicate active microbial communities and viable communities found in plant compartments are distinct. This finding aligns with our flow cytometry data, which indicates only a fraction of microbes present in the plant are active.

## Relative abundance in the rhizosphere viable fraction is associated with relative abundance in the rhizosphere active fraction

For microbial ASVs in the rhizosphere, abundance in the viable fraction was associated with abundance in the active fraction consistently across all replicate pots. (Fig. 5A, $R^2 =$ 0.28, $P < 0.001$, df = 1064). This association between abundance and activity is also

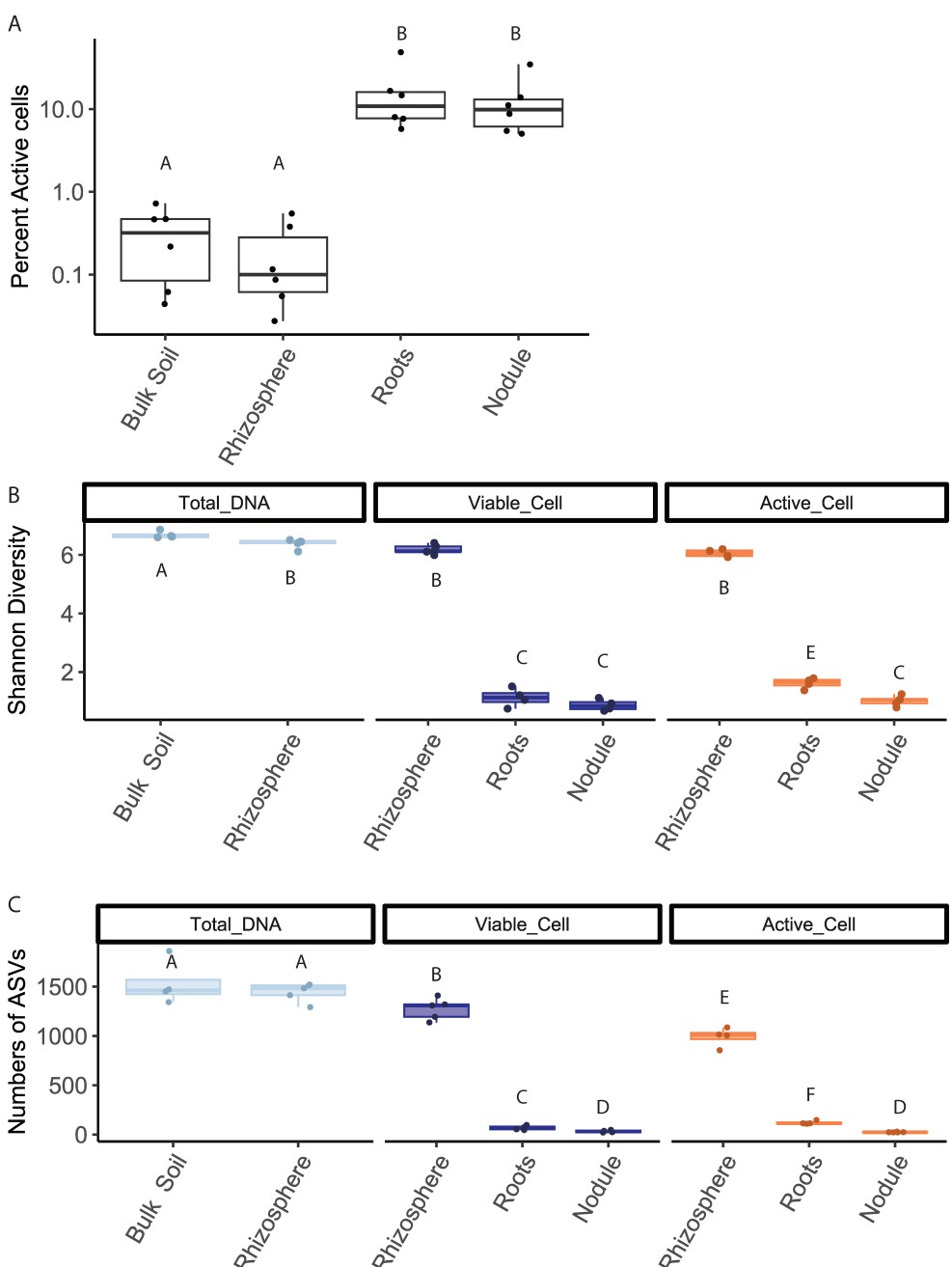

**FIG 3** Microbial activity increases while microbial diversity decreases. (A) Percent (BONCAT) active cells across compartments. Counts of active cells and viable cells were measured on the BD Fortessa flow cytometer. The Y-axis is on a log $_{10}$ scale. Letters indicate significant differences in the portion of active cells to viable cells between compartments (likelihood ratio test, $P >$ 0.05). Replicates are from different plants ($n = 6$). A full table of test results can be found in Table S1. (B) Shannon Diversity metric of rarefied reads. (C) ASV richness from rarefied reads. For B) and C) letters indicate significant differences (ANOVA, $P >$ 0.05). Replicates are from different plants ($n = 4$). All test results are in Tables S3 and S4.

present in the 50 most abundant ASVs in the viable fraction (Fig. 5B). All the top 50 ASVs (33% of the viable fraction) are active, which suggests active microbes may be growing with plant-derived resources.

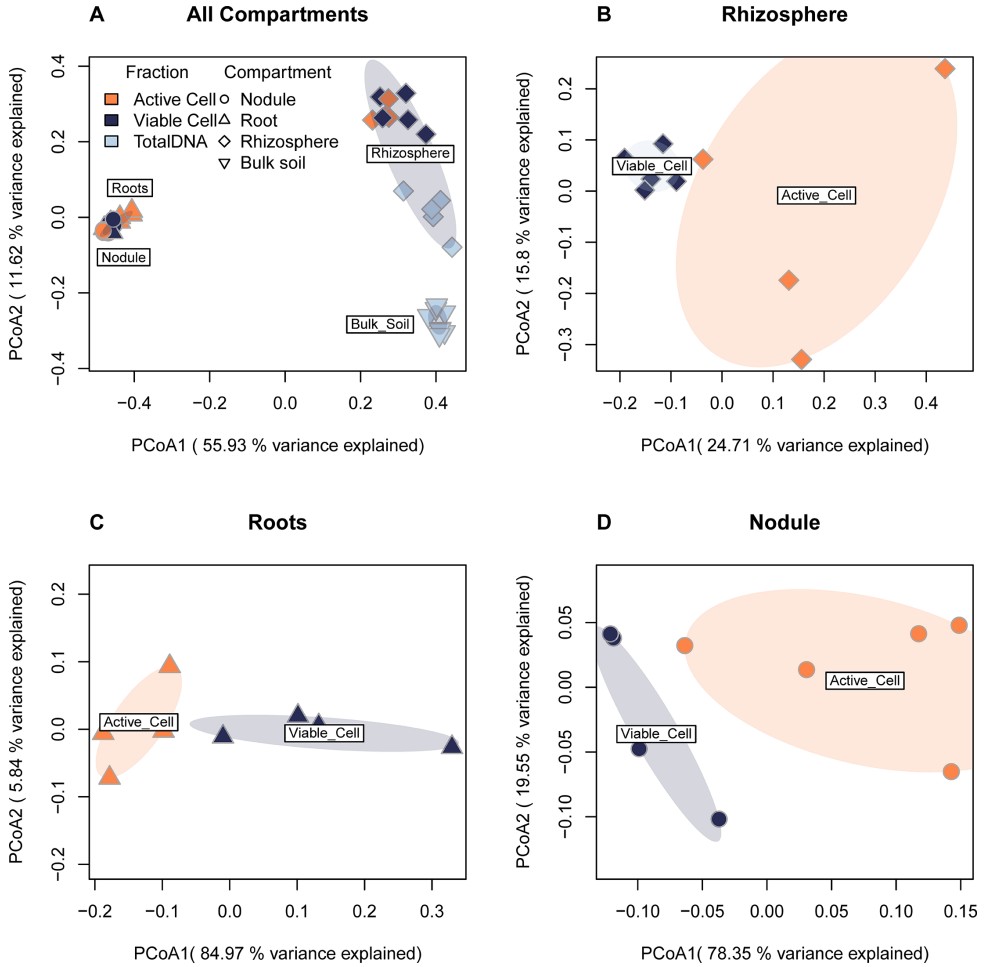

**FIG 4** Active and viable microbial communities are distinct. PCoA of Bray-Curtis dissimilarities of microbial communities on rarefied 16S rRNA gene ASV table. Ovals represent 95% confidence intervals. (A) Total data set showing microbial communities across compartments and fractions. (B) Rhizosphere active and viable communities do not overlap in their 95% CI. In panels C (root endosphere) and D (nodule), a PERMANOVA on Bray Curtis dissimilarity shows distinctions between the viable and active communities. Full beta dispersion and PERMANOVA results are in Tables S6 and S7. Colors indicate the fraction sampled; shapes indicate the compartment sampled.

## There are taxonomic differences between the active and viable communities in the rhizosphere

The relative abundance in the active and viable fractions is statistically different for the top six phyla in our data set (Fig. 6A, from left to right: Proteobacteria, $P < 0.001$, df = 6; Acidobacteria, $P < 0.001$, df = 6; Actinobacteria, $P < 0.001$, df = 6; Verrucomicrobiota, $P < 0.001$, df = 6; Bacteroidota, $P < 0.001$, df = 6; Chloroflexota, $P < 0.001$, df = 6;). Proteobacteria and Acidobacteria are the two most abundant phyla, accounting for more than 60% of the total reads. Both phyla have lower relative abundance in the active fraction than the viable fraction, suggesting that many microbial cells assigned to these phyla are dormant. Conversely, Actinobacteria, Verrucomicrobiota, Bacteroidota, and Chloroflexota have higher relative abundance in the active fraction than the viable fraction. We evaluated if distinct ASVs drove these community differences with differential abundance analysis. In our data set, 48 ASVs are differentially abundant between the active and viable fractions. Fig. 6B highlights the 20 most abundant ASVs with the highest absolute log fold change (full list in Table S8). Several differentially abundant ASVs were enriched in the viable fraction and absent in the active fraction or

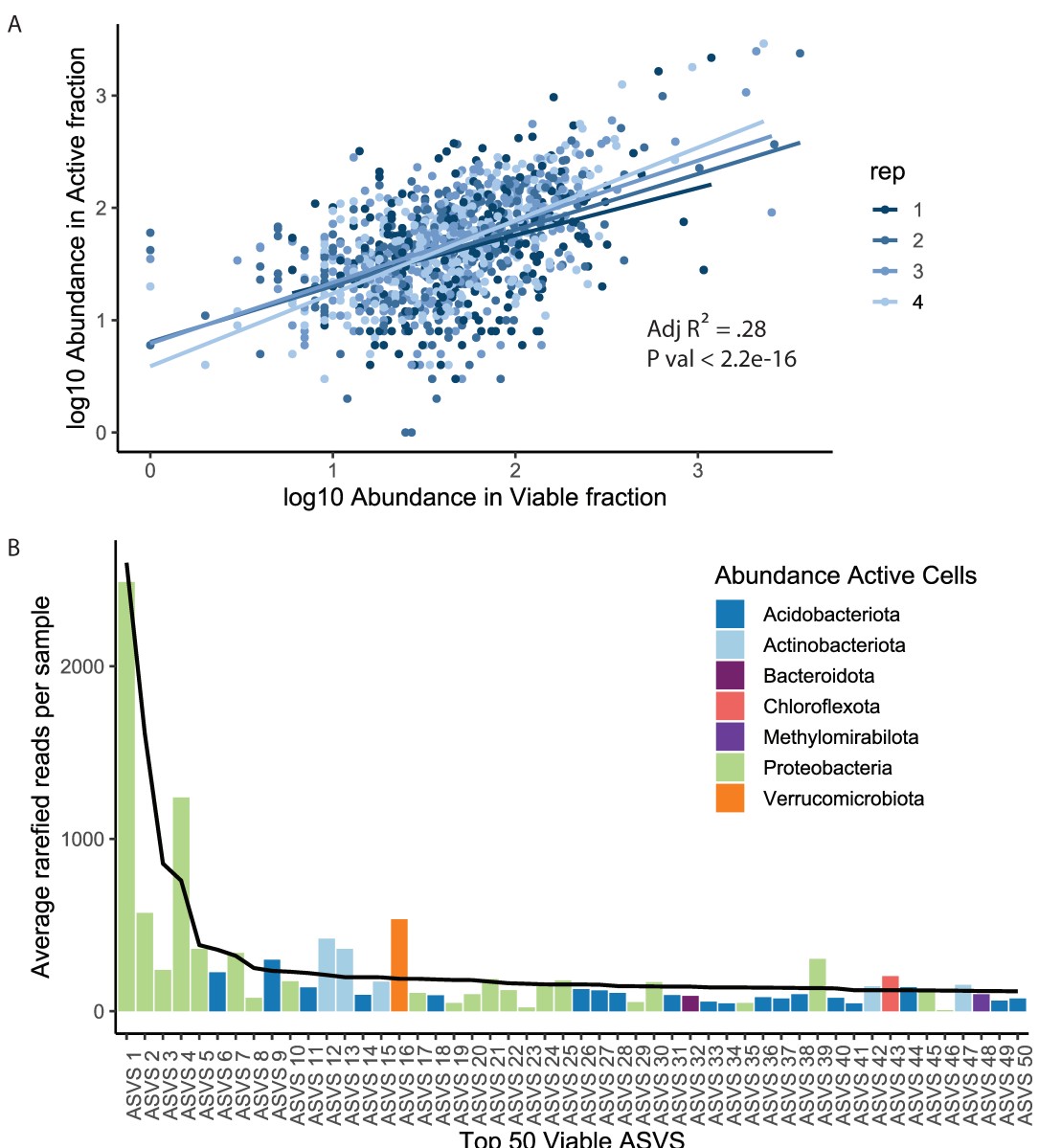

**FIG 5** Relative abundance in the viable fraction is associated with relative abundance in the active fraction.(A) Scatter plots between the $\log_{10}$ abundance of ASVs in the viable fraction and the $\log_{10}$ abundance in the active fraction for the rhizosphere. ASVs were rarefied and filtered for ASVs (i) present in at least three samples in active or viable fraction, (ii) with more than 50 paired reads in the active or viable fraction, and (iii) with nonzero values in the active and viable fraction (log-transformed linear model, $R^2 = 0.28$, df = 1021, $P < 0.001$). The color indicates the replicate plant sampled ($n = 4$). (B) Bar plot of rarefied ASVs average relative abundance ($n = 4$) in the active fraction of the top 50 most abundant ASVs in the rhizosphere viable fraction (accounting for 33% of the viable fraction). Bar plot colors indicate bacterial phyla. The solid line indicates ASV relative abundance in viable fraction.

putatively dormant. Most of these putatively dormant ASVs are from the Proteobacteria phylum and the family Sphingomonadaceae. Most of these putatively dormant ASVs are very rare at less than 1% of the viable fraction; therefore, they are unlikely to create the large change we see at the Phyla level. Conversely, ASVs that are enriched in the active fraction are very abundant in the viable fraction. In the top 25 most abundant ASVs in the viable fraction, 13 ASVs are significantly enriched in the active fraction, meaning they are highly active in the rhizosphere compared to their population size. Many highly active ASVs belong to genera of known plant symbionts like *Rhizobium*, *Pseudomonas*, and *Caulobacter*. Highly active known plant symbionts in the rhizosphere suggest an association between activity in the rhizosphere and utilizing plant resources.

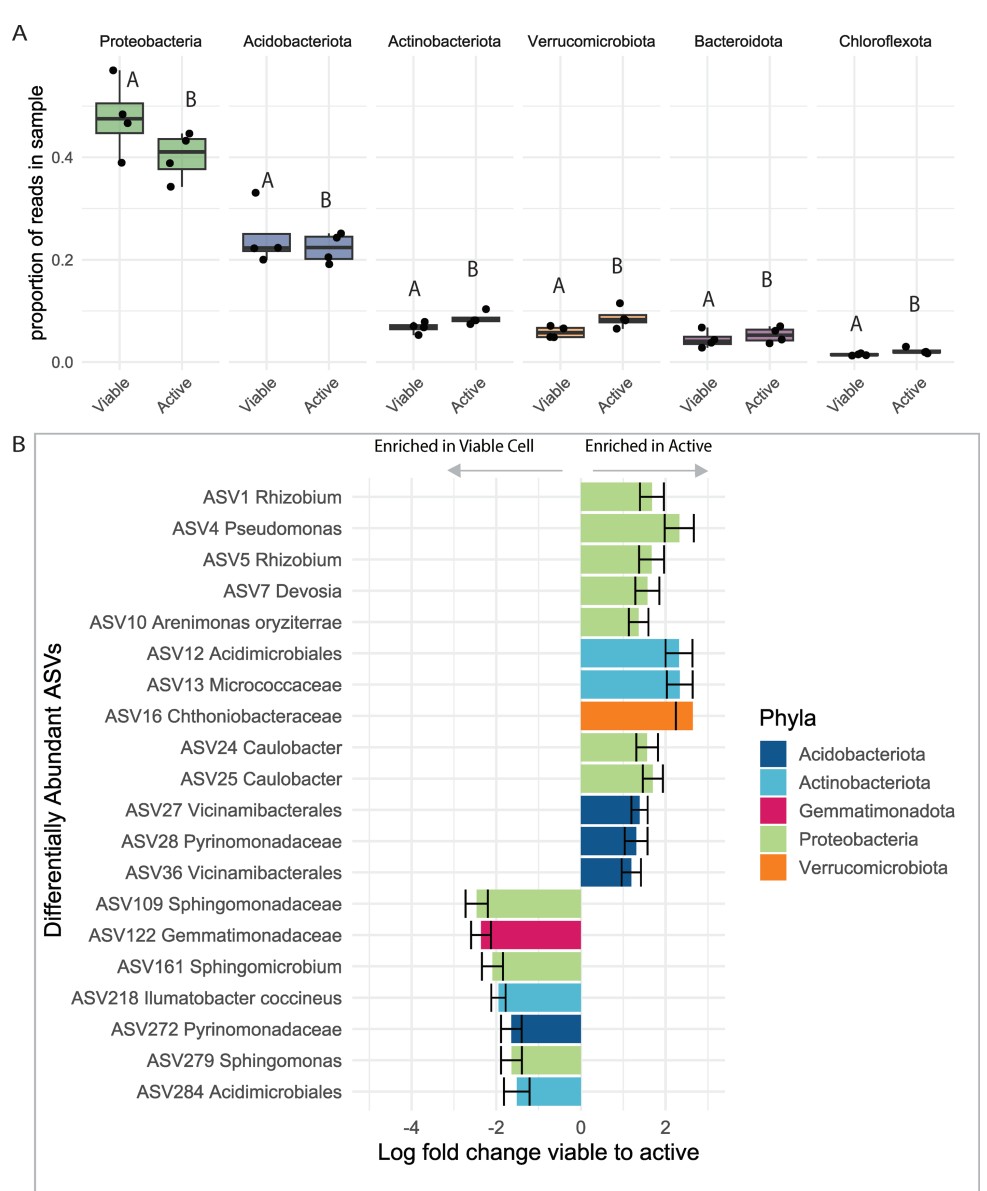

**FIG 6** Differences between the active and viable communities in the rhizosphere propagate from the phylum to the ASV level. (A) Comparison of the relative abundance between viable and active fractions at the phylum level and displaying the top six most abundant phyla. Letters indicate significant differences in the proportion of viable vs active for each phylum (Likelihood ratio test, $P < 0.001$). (B) Bar plot of log fold change (LFC) from viable fraction to active fraction for significantly differentially abundant ASVs (ANCOM, adjusted $P < 0.05$). All ASVs with negative LFC are shown, and the 13 most abundant ASVs in the viable fraction with positive LFC are shown. Error bars are the standard error of LFC. ASVs are labeled by their abundance and their lowest taxonomic identification. For both panels A and B, ASVs were rarefied and filtered for ASVs (i) present in at least three samples in active or viable fraction and (ii) with more than 50 paired reads in the active or viable fraction.

We further explore this by evaluating the relationship between rhizosphere activity and plant colonization.

## Microbial activity in the rhizosphere is associated with plant colonization

Plant below-ground structures exhibited a strong pattern of ecological filtering. Only a fraction of the ASVs detected in the rhizosphere are found in the root endosphere or

nodules. This pattern was consistent for the viable and active fractions (Fig. 7A and B). However, in the viable fraction, only 10.4% of ASVs in the rhizosphere are also present in the root endosphere or nodules. At the same time, in the active fraction, we detected 17.9% of ASVs in the plant, demonstrating that more active ASVs colonize the plant than viable ASVs. We find that both abundance in the rhizosphere viable fraction and abundance in the rhizosphere active fraction are significantly associated with successful plant colonization (binomial model, $P < 0.001$). We used AIC-based model selection to determine whether abundance in the viable fraction or abundance in the active fraction was more effective predictors of plant colonization. We find that including either the abundance in the active (AIC = 555.46, df = 2463) and viable (AIC = 572.01, df = 2463) rhizosphere fractions improved the model relative to the null (AIC = 600.32, df = 2464). However, abundance in the active fraction reduced the AIC by ~17 compared with abundance in the viable fraction, indicating that being active was more important than simply being present (Fig. S8, statistics in Table S8). A few ASVs drive the pattern in our logistic regression; these ASVs are highly abundant in the active fraction in the rhizosphere and present in the plant. The most abundant ASVs contributing to this pattern were from the families of Marinilabiliaceae, Pseudomonadaceae, Caulobacteraceae, Devosiaceae, and Rhizobiaceae, which are in high abundance in the active fraction across compartments. (Fig. 7C, Table S9). Our results suggest that abundant and active ASVs in the rhizosphere are most likely to colonize plant below-ground structures.

## DISCUSSION

We demonstrate that BONCAT is an effective way to label and visualize the active microbial community in plants. A microbe's ability to colonize a plant is typically determined by microscopy of mutant fluorescent microbes, which limits the taxonomic complexity that can be studied and reduces microbes' fitness compared with wild type (53). BONCAT has neither drawback. With BONCAT, we observe labeled endophytic bacteria of the expected size and shape (54) in a complex native community. We expect BONCAT labeling of microbes to function in multiple plant systems because methionine (and therefore the HPG probe, a methionine analog) transporters are conserved across many species (55, 56). Methionine can enter the plant with diffusion or active transport (57–60), and in the plant, methionine can move in the phloem and enter plant cells with membrane-bound transporters (61–63), which is demonstrated by BONCAT-labeled plant cells in our study. Once the HPG probe is in the cell, BONCAT only labels metabolically active cells; previous research shows that metabolically inhibited cells incubated with HPG do not have a BONCAT positive signal (35, 38). Overall, our work opens the door to studying microbial lifestyles, dormancy, and activity within plant tissue.

The rhizosphere is often called a microbial "hotspot" because roots release microbial resources like labile carbon and amino acids (29, 64). Yet, our results show that this microbial "hot spot" depends on what is being measured. The rhizosphere consistently has higher microbial biomass than bulk soils (18, 27, 63, 64), but our data show a microbial biomass hotspot does not equate to a relative microbial activity hotspot. We find 3–4 times more microbial cells in the rhizosphere as compared with the soil (Fig. S4). This aligns with previous research that used PLFA (61) and chloroform extractions (18, 65, 66) to show the rhizosphere has a higher microbial biomass than the bulk soil. Indicators of microbial activity, like RNA-to-DNA ratios (29) and enzymatic activity (18), are higher in the rhizosphere than in the bulk soil. Our results show that in both the rhizosphere and bulk soil, active microbial cells make up ~0.5% of viable cells. Our results align with previous literature when we consider percentage active cells in a relative metric. With the same percentage, higher cell density in the rhizosphere indicates a greater absolute number of active cells in the rhizosphere than the bulk soil (18). In most soils, active microbes compose only about 0.1–2% of the total microbial biomass (17). We used agricultural soils in our study, which typically have lower microbial activity than uncultivated land (67). Therefore, the amount of activity we measured in the soil is in line with the literature, suggesting that the low relative activity in the rhizosphere is not due

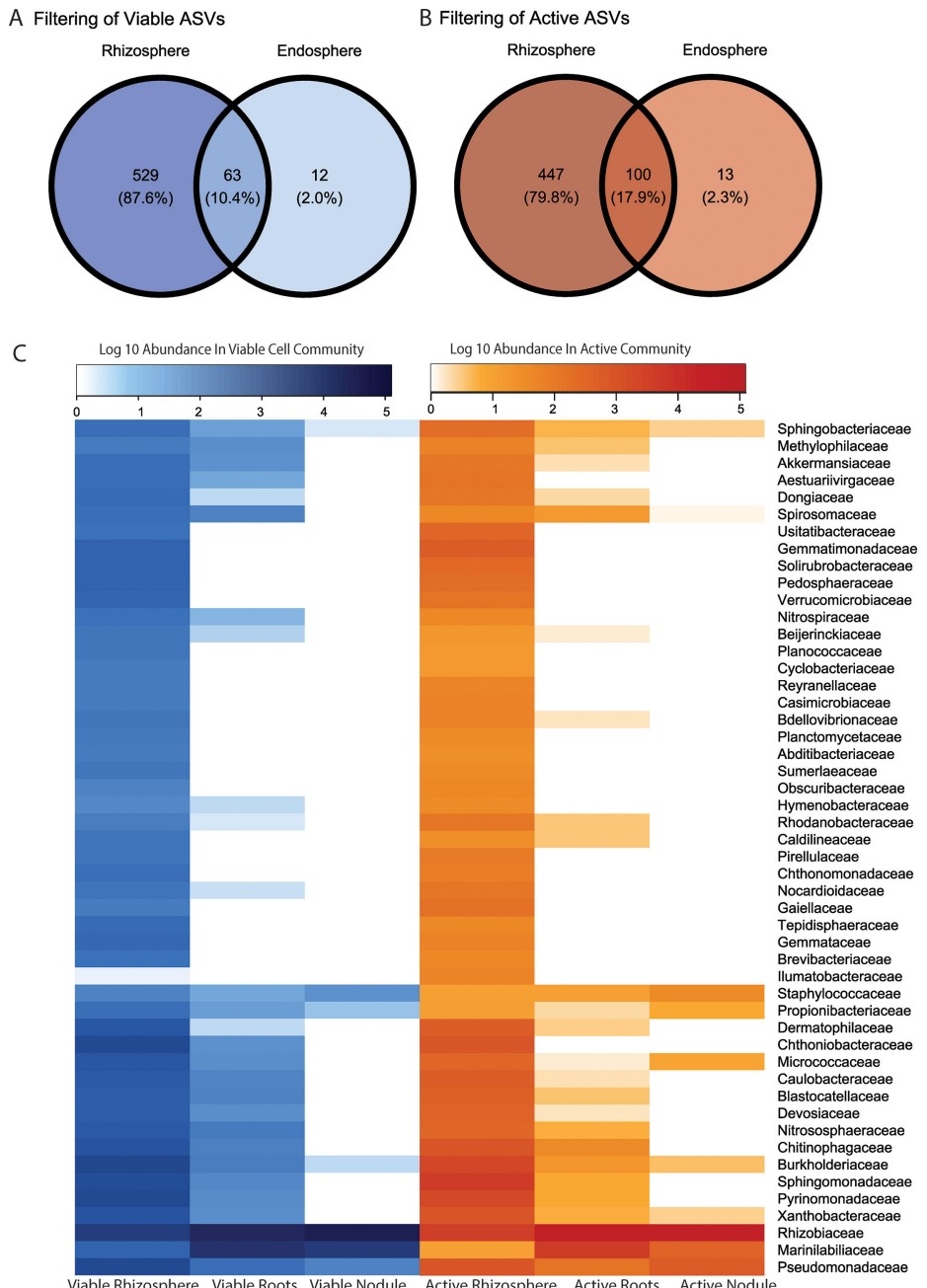

FIG 7 Active taxa are more likely to colonize plants than viable taxa. (A) Venn diagram of the overlap of viable ASVs from the rhizosphere to the endosphere (roots and nodules combined). (B) Venn diagram of the overlap of active ASVs from the rhizosphere to the endosphere (roots and nodules combined). (C) Heatmap of ecological filtering of families from the rhizosphere, roots, and nodules. The 50 most abundant families across all fractions and compartments are shown. Families' abundance in the viable and active microbial fractions is clustered by the ward method. ASVs were rarefied and filtered for ASVs (i) present in at least three samples in active or viable fraction and (ii) with more than 50 paired reads in the active or viable fraction.

to a technical limitation. Likewise, we expect the plant did not out-compete microbes for the HPG probe because previous research has shown soil microbes are better competitors for free methionine than plants (68). Observed activity is sensitive to the sampling scale, sampling timing, and study species. Hot spots of microbial activity are often at the millimeter scale (69), so aggregating the whole root system may make it challenging

to measure activity hotspots. The method of rhizosphere collection affects the results (70), so it is possible our method collected more rhizosphere soil and less rhizoplane, damping the signal of plant-induced microbial activity. Second, root exudates change through the plant's life cycle (71, 72), so microbial activity likely varies over time. Finally, the rhizosphere's effect on the microbiome varies between plant species (24, 73). Our results indicate that the widely held paradigm that rhizosphere increases microbial activity has more nuances than previously described.

We observe strong ecological filtering across the rhizosphere, root, and nodule for the active and viable community. We expect that distance from the root will be inversely proportional to microbial resources (74) and microbial stresses (75). This pattern is reflected in our community, diversity, and activity data. In line with previous work (18, 21, 22), there are clear distinctions in the microbial community between plant tissue, rhizosphere, and bulk soils. There is increasing activity in plant tissues compared with soil, likely driven by abundant plant-produced carbon compounds. However, plant tissue can also be a stressful environment due to the plant's immune system (75–78). We observe a steep drop in active and viable microbial diversity in the plant tissue compared to the rhizosphere, like previous studies on the total DNA community (21, 78, 79). Unexpectedly, we observed that 90% of microbial cells were putatively dormant inside plant tissues, and the active and viable communities differed. Our results suggest endophytic microbes may leverage dormancy to cope with stress, which is a common strategy in other environments (28). The plant immune system can stress endophytes with the production of reactive oxygen species, defense compounds, and toxins (76, 77, 80, 81). After using dormancy to cope with these stresses, endophytes may then exit dormancy when conditions become favorable. Previous work has shown that endophytes can sense environmental cues, like pathogens or abiotic stress, and alter their metabolism to elicit a plant response (82–84). However, in our study, we delivered aqueous HPG into the soil of a living plant, and the probe was transported via the plant's vascular system. With this method, microbes living interstitially that are not exposed to plant resources may not have been labeled, which could create a marginally lower overall percent activity. Nevertheless, our results suggest that widespread microbial dormancy found in other environments like soils, marine environments, and mammalian hosts (85–88) also occurs in plant hosts.

Our study indicated that successful plant endosphere colonization is associated with abundance in the rhizosphere active community. It is expected that abundant taxa would be more successful colonizers because they have more opportunities to establish in a new habitat (89). However, we show abundance in the active community is more important than abundance alone for successful root colonization. Multiple causal mechanisms could underpin the correlations demonstrated by this study. For example, active microbes may be able to more easily access the plant root with chemotaxis, which is an important factor in forming rhizosphere community formation (90). In our data, active ASVs of the genus *Caulobacter* and *Devosia* are successful plant colonizers and have been demonstrated to be capable of chemotaxis (91, 92). Although further work is need to uncover the precise mechanism, our results suggest overcoming dormancy in the rhizosphere is the critical step for plant root colonization.

To be a successful candidate for inoculum, a microbe must be effective *in situ*, competitive in the soil, and colonize the plant (1, 2, 93–96). We leveraged BONCAT to pinpoint microbes that may have an outsized impact on plant health. We focus on unicellular bacterial taxa because filamentous fungi or bacteria cannot be sorted with FACS (97). We found that several ASVs in the rhizosphere were differentially abundant and enriched in the active fraction from genera, such as *Solirubrobacter*, *Capsulimonas*, and *Neorhizobium*. These ASVs, despite their relatively small population size, are likely to have a significant impact on the rhizosphere due to their enrichment in the active community. Our research shows that these last two traits are highly associated. Microbes that are highly active in the rhizosphere are much more likely to colonize the plant compared to their competitors. Some of these microbial taxa have been

well-documented in the literature, such as *Rhizobium* and *Pseudomonas*, which are known to form mutualistic relationships with plants and are commonly found in roots and leaves (23, 98). We also found some less well-described taxa that exhibit similar patterns, including ASVs belonging to the *Caulobacter* genus, which have recently been shown to have a positive effect on plant growth (99). Our findings show that overcoming dormancy is critical for successful establishment; screening the active microbial taxa *in situ* could increase the efficacy of microbial management.

## ACKNOWLEDGMENTS

We acknowledge the Huck Institutes' Flow Cytometry Core Facility (RRID:SCR_024460) for the use of the BD Fortessa Flow Cytometer for data in Fig. 3A and the collection of cells used for 16S rRNA analysis. We acknowledge the Huck Institutes' Metabolomics Core Facility (RRID:SCR_023864) for use of the Exactive Plus LC-MS for data in Fig. S1. We also thank the Penn State Ecology Institute for seed funding.

This work is supported by the USDA National Institute of Food and Agriculture and Hatch Appropriations under Project #PEN04949 and Accession #7006508 to E.C. and Project #PEN04760 and Accession #1025611 to L.T.B. This work is supported by Agricultural Microbiomes in Plant Systems and Natural Resources, Award # 2022-67013-36860 to L.T.B., and NIFA Predoctoral Fellowship, Award # 2023-67011-40517 to J.E.H., both from the U.S. Department of Agriculture's National Institute of Food and Agriculture.

## AUTHOR AFFILIATIONS

[1]Department of Plant Science, The Pennsylvania State University, University Park, Pennsylvania, USA

[2]Ecology Program, Huck Institute for the Life Sciences, The Pennsylvania State University, University Park, Pennsylvania, USA

[3]Department of Ecosystem Science and Management, The Pennsylvania State University, University Park, Pennsylvania, USA

[4]Department of Plant Pathology and Environmental Microbiology, The Pennsylvania State University, University Park, Pennsylvania, USA

## PRESENT ADDRESS

Regina B. Bledsoe, NewLeaf Symbiotics, Inc, Saint Louis, Missouri, USA
Haneen Omari, RETI Center, Brooklyn, New York, USA

## AUTHOR ORCIDs

Jennifer E. Harris http://orcid.org/0000-0003-0615-509X
Liana T. Burghardt http://orcid.org/0000-0002-0239-1071
Estelle Couradeau http://orcid.org/0000-0002-3947-2529

## FUNDING

| Funder | Grant(s) | Author(s) |
| --- | --- | --- |
| National Institute of Food and Agriculture | PEN04949 | Estelle Couradeau |
| National Institute of Food and Agriculture | PEN04760, 2022-67013-36860 | Liana T Burghardt |
| National Institute of Food and Agriculture | 2023-67011-40517 | Jennifer Ellen Harris |

## AUTHOR CONTRIBUTIONS

Jennifer E. Harris, Data curation, Formal analysis, Funding acquisition, Investigation, Methodology, Project administration, Resources, Software, Validation, Visualization, Writing – original draft, Writing – review and editing | Regina B. Bledsoe, Investigation, Methodology, Project administration, Writing – review and editing | Sohini Guha, Methodology, Writing – review and editing | Haneen Omari, Investigation, Methodology, Writing – review and editing | Sharifa G. Crandall, Conceptualization, Funding acquisition, Writing – review and editing | Liana T. Burghardt, Conceptualization, Funding acquisition, Project administration, Supervision, Writing – review and editing | Estelle Couradeau, Conceptualization, Funding acquisition, Investigation, Methodology, Resources, Supervision, Writing – review and editing

## DATA AVAILABILITY

The sequencing data are available at National Center for Biotechnology Information (NCBI) Sequence Read Archive (SRA), under BioProject PRJNA1111431. The flow cytometry and microscopy data are available upon request. The R code for analysis is available at https://github.com/jennnnnharris/.

## ADDITIONAL FILES

The following material is available online.

### Supplemental Material

**Supplemental Material (mSystems00458-25-S0001.pdf).** Supplemental methods, Fig. S1 to S8, and Tables S1 to S10.

### Open Peer Review

**PEER REVIEW HISTORY (review-history.pdf).** An accounting of the reviewer comments and feedback.

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
