## [Reviewer comments · mSystems]

The activity of soil microbial taxa in the rhizosphere predicts the success of root colonization

Jennifer Harris, Regina Bledsoe, Sohini Guha, Haneen Omari, Sharifa Crandall, Liana Burghardt, and Estelle Couradeau

Corresponding Author(s): Estelle Couradeau, The Pennsylvania State University

Review Timeline:

Submission Date:

April 2, 2025

Accepted:

June 24, 2025

Editor: Cheng Gao

Reviewer(s): The reviewers have opted to remain anonymous.

Transaction Report:

DOI: <https://doi.org/10.1128/mSystems.00458-25>

Re: mSystems00458-25 (The activity of soil microbial taxa in the rhizosphere predicts the success of root colonization)

Dear Dr. Estelle Couradeau:

Your manuscript has been accepted, and I am forwarding it to the ASM production staff for publication. Your paper will first be checked to make sure all elements meet the technical requirements. ASM staff will contact you if anything needs to be revised before copyediting and production can begin. Otherwise, you will be notified when your proofs are ready to be viewed.

Cover Image Submissions: If you would like to submit a potential Cover Image, please email a file and a short legend to mSystems@asmusa.org. Please note that we can only consider images that (i) the authors created or own and (ii) have not been previously published. By submitting, you agree that the image can be used under the same terms as the published article. Image File requirements: TIF/EPS, 7.5 inches wide by 8.25 inches tall (at least 2,250 pixels wide by 2,475 pixels tall), minimum 300 dpi resolution (600 dpi preferred), RGB, and no figure elements, e.g., arrows or panel labels. The legend should be a short description of the image, 1-2 sentences recommended. Please download and use this interactive template in Adobe to ensure that your proposed cover image meets our size requirements (<https://journals.asm.org/pb-assets/pdf-text-excel-files/ASM-Interactive-Sizing-Cover-Template-1715689791.pdf>).

Sincerely,
Cheng Gao
Editor
mSystems

Reviewer #6 (Comments for the Author):

The authors have addressed all the concerns.